# Spatio-temporal trends in observed and downscaled precipitation over Ganga Basin

Himanshu Arora<sup>1</sup>, Chandra Shekhar Prasad Ojha<sup>2</sup>, Wouter Buytaert<sup>3</sup>, Gujjanadu Suryaprakash Kaushika<sup>1</sup> and Chetan Sharma<sup>1</sup>

5 <sup>1</sup>Research Scholar, Department of Civil Engineering, Indian Institute of Technology Roorkee, Roorkee, 247667, India <sup>2</sup>Professor, Department of Civil Engineering, Indian Institute of Technology Roorkee, Roorkee, 247667, India <sup>3</sup>Senior Lecturer, Faculty of Engineering, Department of Civil and Environmental Engineering, Imperial College London, U.K.

*Correspondence to*: Himanshu Arora (himanshuuaroraa@gmail.com)

- Abstract. This paper focuses on the spatio-temporal trends of precipitation over the Ganga Basin in India for over 2 10 centuries. Trends in precipitation amounts are detected using observed data for historical period in 20th century and using downscaled precipitation data from 37 GCMs for 21st century. The ranking of 37 GCMs (from CMIP5 archive) is done employing a statistics based skill score. The best ranked GCM output is then bias corrected with observed precipitation prior to further analysis. The direction and magnitude of trend in annual and seasonal precipitation series is determined using
- Mann Kendall's test statistic (ZMK) and Thiel Sen's Slope estimator ( $\beta$ ). The plots depicting the spatial variation of ZMK 15 and  $\beta$  are prepared which provides a comprehensive inter-scenario comparison of spatio-temporal trends in precipitation series. Highly non-uniform spatio-temporal trends are detected for observed precipitation series. It is observed that the precipitation for annual and southwest monsoon season is indicating a rising trend for all future emission scenarios in the region adjacent to Himalayas (northeast side of study area) but shows falling trends in the plains away from the Himalayas.
- 20 Insignificant trends are observed in pre-monsoon and winter season precipitation. An inter-emission-scenario comparison shows that for higher emission scenarios the annual and southwest monsoon precipitation is showing rising trends with increasing spatial dominance i.e. the area under rising trends increases as we observe it from low to high emission scenarios.

## **1** Introduction

The Ganga basin is of great social, economic and religious importance to India. The basin which is a part of northern plains 25 of India is predominantly covered by fertile alluvial soils brought down by various tributaries of Ganga. Agriculture being the dominant economy sector in this basin is therefore dependent on rainfall. The monsoon season for the region is mainly in the months of June to September. Similar is the season for cultivation of Kharif season crops in India. Main Kharif crops are rice and millet, which are highly dependent on quantity and timing of availability of rain water. More than 85% of the total rainfall occurs in monsoon season in this region. Rabi crops are sown in winter and harvested in spring season. In India, the

main Rabi crop is wheat, for which the main source of water is groundwater and irrigation. Excess rainfall in this season may spoil the Rabi crops but on the contrary is beneficial to Kharif crops.

Interpreting the importance of rainfall in the region, this study is targeted for determining the possible future trends in precipitation for the area. Ganga basin is already of great interest to many researchers, hydrologists and water resource

- planners. Few of them have already reported the trends in observed precipitation series in some regions which are subsets of Ganga basin (Kothyari et al. 1997; Duhan and Pandey 2013; Suryavanshi et al. 2013). They have utilized monthly time series of observed precipitation and aggregated them to annual and seasonal amount to capture the trends embedded in them. A number of studies related to trends analysis of rainfall/precipitation are reported in literature. Diversities are found in various studies from around the world which include not only temporal trends but also spatial trends in precipitation patterns,
- out of which some of them are enlisted in Table 1. The studies are based on rainfall/precipitation on various time scales such: annual extreme of 5-30 min and 1-12 h rainfall (Adamowski et al. 2003), annual total rainfall, seasonal total rainfall (as per most of the authors enlisted in Table 1); whereas as in few studies, researchers have focused on rainfall amounts of a particular season only such as monsoon season rainfall (Bashistha et al. 2009). Some authors have also utilized certain indices for trend detection such as frequency indices in daily rainfall for total, light, moderate, intense and very intense
- rainfall (Gallego et al. 2011), seasonality index (Celleri et al. 2007). Most of the researchers have used rain-gauge station data, whereas few of them have considered data for meteorological subdivisions of a region for trend analysis (Guhathakurta and Rajeevan, 2008).

Popular methods used by researchers for trend analysis of rainfall/precipitation time series are parametric linear regression analysis (Guhathakurta and Rajeevan 2008; Caloiero et al. 2011; Kumar et al. 2013), robust locally weighted regression

- (Kothyari et al. 1997), non-parametric Mann-Kendall's test (used by most of the authors enlisted in Table 1), modified Mann-Kendall's Test (Bashistha et al. 2009), Sen's slope estimator (Gallego et al. 2011; Barua et al. 2013; Duhan et al. 2013; Suryavanshi et al. 2013). Jain and Kumar (2012) have shown insignificant Sen's slope estimates for annual as well as seasonal observed rainfall series in Ganga Basin as a whole. Duhan and Pandey (2013) have also observed a dominating negative trend at most of the rain gauge stations except for a few in the study area which lies in western part of Ganga Basin.
- Arora et al. (2016) have provided a scenario-wise comparison of downscaled rainfall over a upper Yamuna sub-basin. A number of agencies have developed their own GCMs following the norms provided by Intergovernmental Panel on Climate Change, yet every GCM cannot be stated as ideal for a particular location. Ranking for best fitting GCMs is done on the basis of skill scores which are subsequently based on certain statistical parameters in order to determine the GCMs with best precipitation capturing capability (Perkins et al., 2007; Ojha et al., 2013)
- In this study, along with the spatial and temporal trends in annual and seasonal precipitation, future-emission-scenario-wise trends are also presented. For this, the observed and downscaled precipitation data is used which is made available by various sources in gridded form. Ranking is done on the basis of skill score which is further based on various statistical parameters and other statistics. Bias correction is performed on downscaled precipitation to reduce the systematic biases present in them. After that the trend analysis is performed on observed and bias-corrected downscaled precipitation series

over the entire study area. The methods used for this purpose are Mann-Kendall's test and Thiel-Sen's slope estimator. The details of study area and methodology followed are discussed in subsequent sections.

Table 1: Summary of studies related to trend analysis of rainfall/precipitation

| No.                      | References and study area                                                                                 | Duration and data length                             | Methods employed                           |  |  |
|--------------------------|-----------------------------------------------------------------------------------------------------------|------------------------------------------------------|--------------------------------------------|--|--|
| Studies around the World |                                                                                                           |                                                      |                                            |  |  |
| 1                        | Lettermaier et al. (1994)                                                                                 | monthly precipitation for data between 1948-         | Nonparametric seasonal Kendall's test,     |  |  |
|                          | 1036 stations in continental United                                                                       | 1988                                                 | Thiel-Sen's slope                          |  |  |
|                          | States                                                                                                    |                                                      |                                            |  |  |
| 2                        | Adamowski et al. (2003)                                                                                   | annual extreme of 5,10, 15, 30 min and 1, 2, 6       | Regional average Mann-Kendall S trend      |  |  |
|                          | 44 rainfall stations in Ontario,                                                                          | and 12 h rainfall with 20 years data                 | test, L moments method, A bootstrap        |  |  |
|                          | Canada                                                                                                    |                                                      | methodology                                |  |  |
| 3                        | Celleri et al. (2007)                                                                                     | seasonality index and;                               | Two sided Mann-Kendall's test              |  |  |
|                          | 23 rainfall stations, Paute Basin,                                                                        | annual, seasonal and monthly rainfall for data       |                                            |  |  |
|                          | Ecuadorian Andes                                                                                          | between 1964-1998                                    |                                            |  |  |
| 4                        | Aldrian et al. (2008)                                                                                     | annual changes in monthly and seasonal               | Empirical Orthogonal Function (based on    |  |  |
|                          | 40 rainfall stations, Brantas                                                                             | rainfall with 50 years data (1955-2005)              | multivariate statistics), non-parametric   |  |  |
|                          | Catchment Area (DAS Brantas), East                                                                        |                                                      | Mann-Kendall Test, Wavelet Transform       |  |  |
|                          | Java                                                                                                      |                                                      | Method                                     |  |  |
| 5                        | Beecham and Chowdhary (2010)                                                                              | point rainfall (rainfall intensities at 0.1, 0.5, 1, | Statistical moments, lag1 autocorrelation, |  |  |
|                          | Melbourne, Australia                                                                                      | 3, 6, 12 h and monthly) for data between             | the Buishand's Q test, Mann-Kendall test   |  |  |
|                          |                                                                                                           | 1925-2002                                            | and wavelet analysis                       |  |  |
| 6                        | Gallego et al. (2011)                                                                                     | frequency indices in daily rainfall (total, light,   | Mann-Kendall test, Sen's slope estimator   |  |  |
|                          | 27 stations in Portugal and Spain,                                                                        | moderate, intense and very intense rainfall)         |                                            |  |  |
|                          | Iberian Peninsula                                                                                         | for data between1903-2003                            |                                            |  |  |
| 7                        | Caloiero et al. (2011)                                                                                    | annual and seasonal rainfall with 50 years           | Mann-Kendall test, parametric linear       |  |  |
|                          | 109 rain gauge stations, Calabria                                                                         | data                                                 | regression analysis                        |  |  |
|                          | (Southern Italy)                                                                                          |                                                      |                                            |  |  |
| 8                        | Iwasaki (2012)                                                                                            | hourly rainfall amounts for June and                 | Non-parametric Wilcoxon rank-sum test      |  |  |
|                          | 758 stations in Eastern Japan                                                                             | September with 31 years data (1976-2006)             |                                            |  |  |
| 9                        | Casimiro et al. (2013)                                                                                    | annual and seasonal rainfall with 40 years           | Mann-Kendall non- parametric test,         |  |  |
|                          | 58 stations in Peruvian Amazon-                                                                           | data                                                 | Pettitt non-parametric test                |  |  |
|                          | Andes basin                                                                                               |                                                      |                                            |  |  |
| 10                       | Barua et al. (2013)                                                                                       | monthly and annual rainfall with 54 years            | Mann-Kendall test, Sen's slope estimator,  |  |  |
|                          | 15 rainfall stations, Yarra River                                                                         | data (1953-2006)                                     | CUSUM test, pre-whitening criteria test    |  |  |
|                          | catchment, Victoria, Australia                                                                            |                                                      |                                            |  |  |
| 11                       | Kumar et al. (2013)                                                                                       | annual, wet and dry seasonal rainfall with 100       | Linear Regression                          |  |  |
| G ( 11                   | 14 stations in Fiji                                                                                       | years data                                           |                                            |  |  |
| Stual                    | es in India<br>Cyliathalizate and Baiagyan (2008)                                                         | monthly, seesand and annual minfall for data         | Linconnection                              |  |  |
| 1                        | 26 metaorological subdivisions of                                                                         | horung, seasonal and annual fannan for data          | Linear regression                          |  |  |
|                          | Jo ineceotological subulvisions of                                                                        | between 1901-2005                                    |                                            |  |  |
| 2                        | muia<br>Gubathakurta at al. (2011)                                                                        | annual one day extreme rainfall with 30 years        | Mann Kandall's test least square linear    |  |  |
| 2                        | 2599 raingauge stations all over India                                                                    | (and more) data                                      | fit                                        |  |  |
| Studi                    | es in Ganga Basin                                                                                         | (and more) data                                      | int                                        |  |  |
| 1                        | 1 Kothyari et al. (1997) total monsoon rainfall with 89 years data robust locally weighted regression. Ma |                                                      |                                            |  |  |
| •                        | 3 stations Agra. Dehradun and Delhi                                                                       | (1901-1989)                                          | Kendall test                               |  |  |
|                          | in Ganga Basin, India                                                                                     | (1)01 1)0))                                          |                                            |  |  |
| 2                        | Bashistha et al. (2009)                                                                                   | annual and monsoon rainfall with 80 years            | Modified Mann-Kendall Test. Pettitt-       |  |  |
| 2                        | 30 rain gauge stations in Indian                                                                          | data (1901-1980)                                     | Mann-Whitney test                          |  |  |
|                          | Himalayas                                                                                                 |                                                      |                                            |  |  |
| 3                        | Duhan et al. (2013)                                                                                       | annual and seasonal precipitation with 102           | Mann-Kendall Test, Thiel-Sen's slope       |  |  |
|                          | 45 stations, Madhya Pradesh, India                                                                        | years data (1901-2002)                               | estimator, Cumulative deviations test and  |  |  |
|                          | , <u>,</u> , <u>,</u>                                                                                     |                                                      | Mann–Whitney–Pettitt method                |  |  |
| 4                        | Suryavanshi et al. (2013)                                                                                 | annual and seasonal precipitation (monsoon,          | Mann-Kendall Test, Thiel-Sen's slope       |  |  |
|                          | Betwa Basin, India                                                                                        | winter and summer)                                   | estimator                                  |  |  |
| 5                        | Arora et al. (2016)                                                                                       | monthly precipitation series                         | scenario-wise comparison                   |  |  |
|                          | Yamuna-Hindon Interbasin, India                                                                           |                                                      |                                            |  |  |

# 2 Study Area and Data Used

## 2.1 Ganga Basin

India lies in a tropical monsoon climate zone with spatially diversified rainfall pattern (Guhathakurta et al. 2011). The study area considered is the portion of Ganga basin which lies in India only. It lies between latitudes from 21.25° N to 31.5° N and

- 5 longitudes from 73.25° E to 89.25° E. As per the Köppen-Geiger climate type map of Asia (Peel et al. 2007), the study area predominantly lies in humid-subtropical climatic zone of India. A smaller region of it in the western most part lies in semiarid zone, whereas the downstream-most part of it lies in tropical wet and dry zone of India. The length of main stream of Ganga is more than 2500 kms and the catchment area is about 20% of the total geographical area of India. The River Ganga originates in the laps of Himalayas (at Gaumukh), flows eastwards through the northern plains of India and finally drains
- 10 into the Bay of Bengal, forming the biggest delta in the world along with the River Brahmaputra. Along its complete stretch, many tributaries unite themselves with the Ganga at various locations. Tributaries like Gomati, Ghaghra and Gandak conjoins with main stream of Ganga from the northern side. These tributaries are glacier fed and are perennial. Whereas tributaries like Chambal, Sind, Betwa, Ken and Son conjoins it from the southern side.