# Peer review of "Spatio-temporal trends in observed and downscaled precipitation over Ganga Basin"

_Hydrology and Earth System Sciences, 2017_

## Referee Comment (RC1) · Anonymous Referee #1 · 19 Jul 2017

Following are my comments that should be answered before this paper is acceptable for publication (1) You mention downscaling of precipitation but there is no text devoted to this subject (2) There is a trend (slope) of the precipitation but I do not see any line plots that show this trend. A trend line would highlight the increase or decrease (3) All statistics should be accompanied with p-value and significance (4) Spatial statistics (standard deviation) should be included in the paper

---

## Referee Comment (RC2) · Anonymous Referee #2 · 12 Aug 2017

There are points that require clarification and further details to be addressed in the manuscript before considering the paper for final publication.

1. Authors used gridded database from observed and CMIP5. But, they used interpolation technique in map. Justification is required about the selection of spatial interpolation method used in preparation of maps.

2. In the Results and Discussion section, authors simply describe the results present in the tables and in the figures, and not discuss those results in depth. The authors should try to improve this section by discussing results in depth and try to compare with other similar works in study area as mentioned in literature part. Also include an interpretation on how the changes will affect e.g. water resources in the study region.

[Figure]

3. Mann-Kendall's test for trend detection (section 3.3.1) is well known and this section can be described in short paragraph (incl. references).

4. Considering the large area of Ganga basin, it is suggested to include the results and discussion based on upper, middle and lower part of Ganga basin.

5. More information and clarity is required in bias correction and downscaling part of methodology and results.

6. Authors should check the references list. There are missing reference of Suryavanshi et al. (2013) Bashistha et al. (2009), Casimiro et al. (2013), Iwasaki (2012), Aldrian et al. (2008), Celleri et al. (2007), Lettermaier et al. (1994).

---

## Author Comment (AC1) · 4 Oct 2017

Reply to Interactive comments from Anonymous Referee #1 (https://doi.org/10.5194/hess-2017-388-RC1)

RC1(1). You mention downscaling of precipitation but there is no text devoted to this subject.

AC1(1). In this study, Authors didn't apply any downscaling procedure, but have utilized the downscaled precipitation data, which is obtained from CMIP5 climate and hydrology projections archive (available at http://gdo-dcp.ucllnl.org/downscaled_cmip_projections/) (Maurer et al. 2007). The precipitation was downscaled (by Reclamation 2014) using the bias-correction and spatial dis-

aggregation (BCSD) method. Downscaled precipitation for 37 GCMs from CMIP5 archive (ensemble r1i1p1) are used in this study. Following reference included in revised manuscript: Reclamation: Downscaled CMIP3 and CMIP5 Climate and Hydrology Projections: Release of Hydrology Projections, Comparison with preceding Information, and Summary of User Needs, prepared by the U.S. Department of the Interior, Bureau of Reclamation, Technical Services Center, Denver, Colorado. 110 pp, 2014

RC1(2). There is a trend (slope) of the precipitation but I do not see any line plots that show this trend. A trend line would highlight the increase or decrease

AC1(2). The plots showing trend lines at particular grid points are added in the revised manuscript.

RC1(3). All statistics should be accompanied with p-value and significance.

AC1(3). The p-values associated with all statistics (such as ZMK) are included in the revised manuscript.

RC1(4). Spatial statistics (standard deviation) should be included in the paper.

AC1(4). The plots showing the spatial variation of standard deviation (similar to spatial plots for mean which are already there in manuscript) are included in the revised manuscript.

---

## Author Comment (AC2) · 4 Oct 2017

Reply to Interactive comments from Anonymous Referee #2 (https://doi.org/10.5194/hess-2017-388-RC2)

RC2(1). Authors used gridded database from observed and CMIP5. But, they used interpolation technique in map. Justification is required about the selection of spatial interpolation method used in preparation of maps

AC2(1). The Authors did not use any interpolation technique in maps. The maps are prepared using gridded data and are presented in gridded form only.

RC2(2). In the Results and Discussion section, authors simply describe the results present in the tables and in the figures, and not discuss those results in depth. The

authors should try to improve this section by discussing results in depth and try to compare with other similar works in study area as mentioned in literature part. Also include an interpretation on how the changes will affect e.g. water resources in the study region.

AC2(2). As per the suggestion, Results and Discussion section have been improved and the necessary changes have been incorporated in the revised manuscript. This comparison of the results with other similar studies for observed data have been included in this section of revised manuscript. To the best of our knowledge, there is no study which deals with the trend analysis of precipitation in 21st century using CMIP5 data in this study area. Thus, a comparison with other studies has not been feasible for future scenarios. It is obvious that increase in precipitation, particularly in Himalayan region will lead to more water yield. In the companion paper for this issue (Shukla et al., 2017), the water yield is found to increase with increasing precipitation in Upper Ganges basin. However, to project such changes in water yield will also require knowledge of changing LULC, which is very uncertain to predict at this stage. For this reason, no discussion about the future status of water resources in Ganges basin was addressed in the paper. However, as advised by reviewer, we have included a part of reply in the revised manuscript also.

RC2(3). Mann-Kendall's test for trend detection (section 3.3.1) is well known and this section can be described in short paragraph (incl. references).

AC2(3). As per the suggestion from the reviewers, the section 3.3.1 is shortened in the revised manuscript.

RC2(4). Considering the large area of Ganga basin, it is suggested to include the results and discussion based on upper, middle and lower part of Ganga basin.

AC2(4). As per the suggestion from referee#2, the study area is being divided in four parts, namely Yamuna basin (YB), Chambal basin (CB), Upper Ganga (UGB) and Lower Ganga basin (LGB). The results are included partwise in the revised manuscript.

RC2(5). More information and clarity is required in bias correction and downscaling part of methodology and results.

AC2(5). In this study, Authors didn't apply any downscaling procedure, but have utilized the downscaled precipitation data, which is obtained from CMIP5 climate and hydrology projections archive (available at http://gdo-dcp.ucllnl.org/downscaled_cmip_projections/) (Maurer et al. 2007). The precipitation was downscaled (by Reclamation 2014) using the bias-correction and spatial disaggregation (BCSD) method. Downscaled precipitations for 37 GCMs from CMIP5 archive (ensemble r1i1p1) are used in this study. In case of bias correction, the steps followed are included in the revised manuscript with relevant references. Reference included in revised manuscript: Reclamation: Downscaled CMIP3 and CMIP5 Climate and Hydrology Projections: Release of Hydrology Projections, Comparison with preceding Information, and Summary of User Needs, prepared by the U.S. Department of the Interior, Bureau of Reclamation, Technical Services Center, Denver, Colorado. 110 pp, 2014

RC2(6). Authors should check the references list. There are missing reference of Suryavanshi et al. (2013) Bashistha et al. (2009), Casimiro et al. (2013), Iwasaki (2012), Aldrian et al. (2008), Celleri et al. (2007), Lettermaier et al. (1994).

AC2(6). The above mentioned references are included in revised manuscript. Bashistha et al. (2009) is corrected to Basistha et al. (2009). Lettermaier et al. (1994) is corrected to Lettenmaier et al. (1994).